# Functional Regulation of ZnAl-LDHs and Mechanism of Photocatalytic Reduction of CO_2_: A DFT Study

**DOI:** 10.3390/molecules28020738

**Published:** 2023-01-11

**Authors:** Dongcun Xu, Gang Fu, Zhongming Li, Wenqing Zhen, Hongyi Wang, Meiling Liu, Jianmin Sun, Jiaxu Zhang, Li Yang

**Affiliations:** 1MIIT Key Laboratory of Critical Materials Technology for New Energy Conversion and Storage, School of Chemistry and Chemical Engineering, State Key Laboratory of Urban Water Resource and Environment, Harbin Institute of Technology, Harbin 150001, China; 2Xupai Power Co., Ltd., Suqian 223800, China

**Keywords:** ZnAl-LDHs, CO_2_PR, DFT, reaction mechanism, defect engineering, Cu doping

## Abstract

Defect engineering and heteroatom doping can significantly enhance the activity of zinc-aluminum layered double hydroxides (ZnAl-LDHs) in photocatalytic CO_2_ reduction to fuel. However, the in-depth understanding of the associated intrinsic mechanisms is limited. Herein, we systematically investigated Zn vacancies (V_Zn_), oxygen vacancies (V_O_), and Cu doping on the geometry and electronic structure of ZnAl-LDH using density functional theory (DFT). We also revealed the related reaction mechanism. The results reveal the concerted roles of V_O_, V_Zn_, and doped-Cu facilitate the formation of the unsaturated metal complexes (Zn^δ+^-V_O_ and Cu^δ+^-V_O_). They can localize the charge density distribution, function as new active centers, and form the intermediate band. Simultaneously, the intermediate band of functionalized ZnAl-LDHs narrows the band gap and lowers the band edge location. Therefore, it can broaden the absorption range of light and improve the selectivity of CO. Additionally, the unsaturated metal complex lowers the Gibbs free energy barrier for effective CO_2_ activation by bringing the d-band center level closer to the Fermi level. The work provided guidance for developing LDH photocatalysts with high activity and selectivity.

## 1. Introduction

CO_2_ gas is widely used as a common chemical in people’s lives, and with mankind’s dependence on fossil fuels, the huge emissions of CO_2_ have led to many climate problems such as the “greenhouse effect” [1,2,3,4]. Recently, CO_2_ gas capture and conversion technologies have become a hot topic of research, and solar-light-driven CO_2_ conversion is a potential strategy for the production of sustainable fuels such as CO, HCOOH, and CH_4_. However, the efficiency is severely limited by the high inertness of the CO_2_ molecule and the inherent limitations of semiconductor photocatalysts, i.e., fast electron-hole recombination and no reaction sites [4,5]. Layered double hydroxides (LDHs) [6,7] have received increasing attention as a potential two-dimensional (2D) photocatalyst because of their mutable metal cation composition and interlaminar anions, which allows for both electronic structure tuning and bandgap manipulation [8,9,10,11,12,13]. The general formula of LDHs is [M^2+^_1−x_M^3+^_x_(OH)_2_]^x+^(A^n−^_x/n_)·zH_2_O, where M^2+^, M^3+^, and A^n−^ represent divalent, trivalent cations, and charge-balancing anions, respectively. By adjusting the M^2+^ and M^3+^ cations, or the LDHs’ size, the energy band structure [14,15,16] can be altered, which can improve the absorption of visible light, and generally does not introduce additional carrier recombination centers [17,18,19]. However, the photocatalytic efficacy of bulk LDHs is inhibited due to their limited exposed surface, poor light absorption, inefficient adsorption and separation of gas, and sluggish photo-induced charge transferability [4,13,15]. Various strategies such as element doping, defect engineering, and heterostructure engineering have been employed to improve the performances of these LDHs [20,21,22,23].

Zhang et al. synthesized ZnAl-LDH nanosheets (u-LDH) with oxygen vacancies (V_O_), and they demonstrated that the introduction of V_O_ caused the unsaturated Zn^+^ and Zn^+^-V_O_ complexes to form, which can facilitate the separation and transfer of photogenerated carriers and thereby increase the surface charge density for efficiently adsorbing and activating CO_2_ molecules [24]. They further synthesized ultrathin ZnAl-LDH nanosheets with doped electron-rich Cu^δ+^ (Cu-u-LDH) and found that V_O_ and Cu^δ+^ with electron-rich properties tremendously promoted the efficiencies of the separation and transfer of photogenerated electrons/holes, N_2_ adsorption, and the activation and reduction of N_2_ to NH_3_ [25]. To screen out better performance photocatalysts, Zhao et al. synthesized a series of V_O_-rich LDH photocatalysts with M^II^ and M^III^ (M^II^ = Mg, Zn, Ni, Cu; M^III^ = Al, Cr), and revealed that Cu-containing LDHs among these M^II^M^III^-LDHs had excellent photocatalytic activity due to the distorted structures, which enhanced chemical adsorption and activation of N_2_ [26]. In 2022, Song et al. combined ultrathin NiMn-LDH nanosheets and metal–organic framework (MOF) structures to form MIL-100@NiMn-LDH, which had a large specific exposed surface area serving as active sites to promote absorbing and activating CO_2_. Its abundant coordination of oxygen vacancies and unsaturated metal sites facilitated the separating/transporting of photogenerated electrons/holes, exhibiting excellent photocatalytic activity and selectivity [27].

Although the above studies have demonstrated that the introduction of transition metal Cu and defects, especially metal vacancies (V_M_) and oxygen vacancies into ZnAl-LDH can efficiently modulate the catalytic activity of LDHs for photocatalytic CO_2_ reduction reaction (CO_2_PR) [28,29,30], it is still unclear how the doping Cu and vacancies impact the structure-activity connection of LDHs materials. Most interestingly, understanding the coordinative effects of CO_2_PR is still limited. In this paper, the crystal structure, electronic structure, and energy band structure of defect-free zinc-aluminum LDH (ZnAl-LDH), oxygen vacancy and zinc vacancy-rich zinc-aluminum LDH (V_Zn_-ZnAl-LDH), and copper-doped defect-rich zinc-aluminum LDH (Cu-V_Zn_-ZnAl-LDH) were calculated by density functional theory (DFT) with van der Waals corrections. The CO_2_PR and HER reactions of the associated materials were computed to highlight the impacts of V_Zn_, V_O_, and Cu doping and their coordinative effects on the photocatalytic CO_2_ reduction activity of ZnAl-LDHs. This work provides a deep understanding of the intrinsic natures for the improved performances of ZnAl-LDHs by introducing vacancies and transition metal doping and a validation of their potential applications for CO_2_ reduction.

## 2. Results and Discussions

### 2.1. Structure Construction of the Functionalized ZnAl-LDHs

The rationality and reliability of the structure of pristine ZnAl-LDH and the computation method are vital since they are the basis for the further calculation of the modified ZnAl-LDHs. The pristine ZnAl-LDH model is developed based on previous reports, and the optimized structure and lattice parameters are shown in Figure 1 [31,32,33,34]. A 3 × 3 × 1 supercell is adopted for ZnAl-LDH according to the position of the characteristic diffraction peaks (110) and (003) of LDHs powder X-ray diffraction (XRD) experimentally, [31] and the obtained unit lattice parameters are (a = b= 3.08 Å, c= 7.75 Å). The Zn:Al ratio of the supercell is 2:1, six Zn atoms surround one Al atom to form an octahedral structure, and the intercalated anions are nitrate anions. A 15 Å vacuum layer is constructed along the z-direction for eliminating the interactions between duplicating plates [35,36,37,38].

The geometrical parameters are optimized by utilizing the PBE and PBE + vdW (DFT + D3) approaches and are compared with the available values in the experiment and theory [24,31,39,40] to screen out the most reasonable method and model. As shown in Appendix A, the PBE method overestimates the lattice parameters since non-bonding interactions are not taken into account. With van der Waals correction included, the geometries given by PBE with the DFT + D3 method are in line with the experimental observations with a maximum difference of 0.028 Å in the Al-O bond, as shown in Appendix A, and PBE + vdW is thus chosen as an appropriate method for the following calculations. To rationalize the stability of the ZnAl-LDH structure, the energy change versus the variation of the lattice parameters is plotted as shown in Appendix A, suggesting the lowest energy point is the one with the experimental lattice parameter [34]. As shown in Figure 2c, the calculated cohesive and surface energies are −1.35 eV/atom and −1.34 eV/Å^2^, respectively, further confirming the surface stability of ZnAl-LDH. The AIMD results as plotted in Appendix A show that the ZnAl-LDH can maintain its original structure at 600 K with the energy change within 0.001 eV, again indicating its excellent structural stability and validity of the model and calculation method.

It is found that the geometrical parameters of the NO_3_^−^ containing ZnAl-LDH are identical to those of the NO_3_^−^ free one, as presented in Figure 1. Additionally, since the surface is the site of the catalytic reduction occurrence, and the interlayer anions are not involved in the reaction, a single-layer NO_3_^−^ free hydrotalcite is used in this paper to reduce the calculation consumption. As shown in Figure 2, the structures of ZnAl-LDH, V_Zn_-ZnAl-LDH and Cu-V_Zn_-ZnAl-LDH were constructed and optimized, respectively. It should be noted that the presence of defects may cause enhanced surface polarization while the spin polarization is rectified by the spin polarization. The Cu atoms in Cu-V_Zn_-ZnAl-LDH are thus corrected by the Hubbard correction (DFT + U) approach [41,42] for considering the interaction of highly correlated electrons of transition metals. Herein, we systematically study how Cu and V_Zn_ affect the electronic behavior, d-band center, and Gibbs free energy barrier of Cu-V_Zn_-ZnAl-LDHs by the DFT + U method.

The optimized key lengths of ZnAl-LDH, V_Zn_-ZnAl-LDH, and Cu-V_Zn_-ZnAl-LDH are compared with the ones measured by the extended X-ray absorption fine structure spectra (EXAFS), as shown in Figure 2a [25,31]. Generally, the Zn-Zn, Zn-O, and Cu-O bond distances are very consistent with the experimental results with differences less than 0.02 Å, indicating the rationality of our structural models. The cohesion, surface, and defect formation energies of the three ZnAl-LDHs are calculated as shown in Figure 2b-d and are all increased in the order of Cu-V_Zn_-ZnAl-LDH < V_Zn_-ZnAl-LDH < ZnAl-LDH, demonstrating that the zinc and oxygen vacancies can stabilize ZnAl-LDH and doping Cu can further improve the stability of ZnAl-LDH. With the stepwise modification of ZnAl-LDH, the Zn-O bond is decreased and the crystal structure significantly distorts. The Zn-O bond length for V_Zn_-ZnAl-LDH decreases to 2.06 Å compared to 2.08 Å for pristine ZnAl-LDH, the shorter bonds result in larger defect ranges and new active centers, facilitating the adsorption of CO_2_. The activity of photocatalytic processes is also improved by structural distortion since more active defect sites can be created on the surface. This is more obvious for Cu-V_Zn_-ZnAl-LDH, where the Cu-O bond length on the defect site is further shortened to 1.97 Å. It could be because the 3d orbital of the Cu is active and in favor of transporting electrons, and thus, aside from defect sites in Cu-V_Zn_-ZnAl-LDH, the largely exposed Cu atoms serve as another new active site for photocatalytic CO_2_ reduction. The unsaturated Zn^δ+^ and Cu^δ+^ are produced with the introduction of Zn vacancies and doping Cu, which caused structural distortions and the formation of Zn^δ+^-V_O_/Cu^δ+^-V_O_ complexes, as shown in Figure 2a. These complexes act as the reaction’s active sites to improve the catalytic activity.

### 2.2. Charge Properties of Functionalized ZnAl-LDHs

Due to the importance of the surface charge distribution for the reaction, the charge properties of the three modified ZnAl-LDHs are calculated. As shown in Figure 3, by inserting Zn defects and dopant Cu to ZnAl-LDH, the charge density increases dramatically and the charge on the surface is mainly localized around Zn^δ+^/Cu^δ+^, facilitating the electron transport, and Cu^δ+^ is enriched with more charge than Zn^δ+^. Furthermore, the unsaturated metal position can effectively accept electrons from the reactant molecules (CO_2_ or H_2_O) and activate them to form relevant intermediates, promoting the charge exchange reaction.

To analyze the charge and bonding situation, the electron localization function (ELF) is calculated on the (001) surface of three ZnAl-LDHs, as shown in Figure 4a, where red, green, and blue indicate high, moderate, and low charge densities, respectively, and the greater the connection between a metal and the OH group and the deeper the color, the better the capacity of a metal to gain or lose electrons. The charge density is gradually increased with the involvement of V_O_, V_Zn_, and the transition metal Cu, indicating that the unsaturated metal atoms can concentrate the surface charge around them and absorb and activate reactants more effectively. The electron density difference (EDD) and Bader charges of the three ZnAl-LDHs as presented in Figure 4b show that the Zn atom in V_Zn_-ZnAl-LDH loses −0.17 e and the Cu atom in Cu-V_Zn_-ZnAl-LDH loses more electrons (−0.55 e), which are more likely to accumulate on the OH group near the defect. Simultaneously, the charge density of oxygen around the unsaturated metal on the defects is enhanced from +1.36 e to +1.42 e, and Cu^δ+^ (+1.45 e) is more pronounced than Zn^δ+^, suggesting the more efficient charge transfer in defected ZnAl-LDHs. Importantly, the DFT calculations show that when V_Zn_ and V_O_ are introduced, a Zn^δ+^-O bond forms between the unsaturated metal Zn^δ+^ on the defects and O, and similarly, a Cu^δ+^-O bond forms with the addition of doping Cu. The formation of the Zn^δ+^-V_O_/Cu^δ+^-V_O_ complex promotes the charge transfer around the defects and their shortened bond lengths expand the defect ranges, resulting in the new active centers for CO_2_ and H_2_O adsorption.

### 2.3. Photocatalytic Activity of Functionalized ZnAl-LDHs

To explore the photocatalytic activity of the functionalized ZnAl-LDHs, the energy band structure, charge density of the valance band maximum (VBM) and conduction band minimum (CBM), and density of states (DOS) of the relevant ZnAl-LDHs are calculated, as presented in Figure 5. The computed energy band gap (Eg) of ZnAl-LDH calculated by the HSE06 method is 3.20 eV, which is consistent with the experimental value of 3.18 eV. As shown in Figure 5a, the Eg dramatically drops to 2.67 and 1.69 eV for V_Zn_-ZnAl-LDH and Cu-V_Zn_-ZnAl-LDH compared to pristine ZnAl-LDH, suggesting the significant improvement of the efficiency of the available light usage and photoexcited electrons transported from VB to CB. This is due to the formation of the intermediate bands by the introduction of defects and transition metals. The intermediate bands split the original Eg into two parts, and the reduction of the energy band gap suppresses the recombination of photogenerated electrons/holes and thus promotes carrier separation and migration. The charge density of VBM and CBM in Figure 5b shows that the introduction of defects and Cu doping allows surface electrons to be more effectively concentrated around the unsaturated Zn^δ+^/Cu^δ+^ after excitation, potentially promoting carrier migration and facilitating the surface catalytic reaction. The new intermediate band, known as defect energy level, is formed by Zn-4s and Cu-3d orbitals, respectively, as shown in Figure 5c. Interestingly, it is near the Fermi level in the forbidden band of V_Zn_-ZnAl-LDH and Cu-V_Zn_-ZnAl-LDH, which favors the acceptance of the electrons excited on VB and then transits to the Zn-3d orbital on CB. The existence of the defect level favors the reduction of excitation energy and improves the efficiency of electron transfer.

The work functions (WF) of the three ZnAl-LDHs are computed to explore the charge transfer. As shown in Figure 6a, the WF value gradually increases with the subsequent addition of V_Zn_, V_O_, and Cu to ZnAl-LDH, and electrons would flow from ZnAl-LDH to the region containing unsaturated Zn or Cu until the Fermi energy level remains constant. The difference in work functions between ZnAl-LDH and V_Zn_-ZnAl-LDH/Cu-V_Zn_-ZnAl-LDH results in a built-in electric field on the perfect surface defect ZnAl-LDH, which tends to enrich more electrons on the surface and thus promotes the surface catalytic reaction, as presented in Figure 6b.

### 2.4. Mechanism of Functionalized ZnAl-LDHs Photocatalytic Reduction of CO_2_

To further understand the photocatalytic CO_2_ reduction reaction and its competitive water-splitting reaction, the adsorption energies of CO_2_, H_2_O, and the product CO on the three ZnAl-LDHs are calculated with the corresponding structures, as shown in Figure 7. The adsorption energies for Cu-V_Zn_-ZnAl-LDH absorbing CO_2_ (−1.81 eV) and H_2_O (−1.21 eV) are the highest, but the lowest for absorbing CO (−0.14 eV), followed by V_Zn_-ZnAl-LDH. The oxygen vacancies and zinc vacancies on the ZnAl-LDH surface can effectively capture CO_2_ and H_2_O molecules, which may be due to the electron-rich effect of the 4s orbital of Zn. In addition, the doping of Cu on top of oxygen vacancies and zinc vacancies forms unsaturated Cu^δ+^, which can capture H_2_O and CO_2_ molecules more efficiently due to the richer electrons in the 3d orbital of Cu. This suggests that defects and Cu doping can significantly increase the activities of adsorbing reactants and quick desorbing products from the surface, thus facilitating the reduction reaction.

It is well known that photocatalytic reduction of CO_2_ and water splitting are a group of competitive reactions. To screen out ZnAl-LDHs photocatalysts with high selectivity to CO_2_PR, the Gibbs free energy diagrams of photocatalytic CO_2_ and water splitting on ZnAl-LDH, V_Zn_-ZnAl-LDH, and Cu-V_Zn_-Zn-Al-LDH as well as the associated thermodynamic reaction pathways are calculated, as shown in Figure 8a–c. The *CO and *H_2_ formation barriers with values of 1.25 and 1.07 eV, respectively, are the lowest and highest for Cu-V_Zn_-ZnAl-LDH among the three ZnAl-LDHs, suggesting its high selectivity for CO_2_ reduction to CO and inhibition of H_2_ evolution. For V_Zn_-ZnAl-LDH, the barrier for CO_2_ conversion is also very low (ΔG = 1.41 eV); thus, CO is the primary reaction product for both defective ZnAl-LDHs. The results conclusively show that Cu^δ+^ or Zn^δ+^ in ZnAl-LDHs can govern the strengths of CO_2_ and H_2_O adsorption as well as those intermediates generated by their photoreduction, and hence determine the selectivity of photocatalytic CO_2_ reduction.

The d-band center (ε_d_) theory [34,35,36,37,38,39] can be employed to clarify the reasons why Cu^δ+^ and Zn^δ+^ cations can strongly control the adsorption of CO_2_ and H_2_O and their photocatalytic conversion. When ε_d_ moves closer to the Fermi level, the influences on the adsorbate become stronger, facilitating electron injection from the surface to adsorbed gases. As illustrated in Figure 9a, by the introduction of V_Zn_, V_O_, and Cu, respectively, the εd gradually approaches the Fermi level from −4.40 to −2.31 eV, indicating that the electron-donating capacity is dramatically enhanced. Therefore, the reaction energy barriers are reduced and the CO_2_ adsorption and activation ability are significantly improved due to the presence of unsaturated Cu^δ+^/Zn^δ+^.

It is well known that semiconductors as CO_2_ reduction photocatalysts must have suitable band edge positions to match the reduction potential of CO_2_/hydrocarbons. The CBM of photocatalysts is more negative, the driving force behind the photocatalytic reaction is stronger and thus the capability of the photocatalyst to reduce CO_2_ will be stronger, as presented in Figure 9b. The standard redox potentials of CBM for ZnAl-LDHs, V_Zn_-ZnAl-LDH, and Cu-V_Zn_-ZnAl-LDH are −0.75, −0.86, and −0.93 V, respectively, and still more than that of CO_2_/CO (−0.53 V), which is sufficient to drive photocatalytic CO_2_PR. Therefore, all three ZnAl-LDHs are evidenced to be thermodynamically favorable for photocatalytic CO_2_ reduction to CO. Importantly, when V_Zn_ and Cu are introduced to ZnAl-LDH in turn, the CBM grows more and more negative from −0.75, −0.86, and −0.93 V, indicating the strengthened capacity for CO_2_ reduction in the order of ZnAl-LDH < V_Zn_-ZnAl-LDH < Cu-V_Zn_-ZnAl-LDH. The heightened performance of ZnAl-LDHs could owe to the following reasons. Firstly, the coordinative effects caused by oxygen and zinc defects in V_Zn_-ZnAl-LDH have been verified by the fact that it has a significantly improved CBM potential compared to pristine ZnAl-LDH. The introduction of V_O_ and V_Zn_ increases the electron aggregation and transfer efficiency of Zn atoms in defected ZnAl-LDH, and the resulting negative CBM potential dramatically enhances the reduction, and thus the photocatalytic ability. Secondly, with copper incorporation, Cu-V_Zn_-ZnAl-LDH exhibits the smallest Eg, which is capable of significantly improving the transmission ability of photogenerated electrons from VBM to CBM, and the energy of CO_2_ adsorption brought on by the high dispersion of Cu-3d atomic electron orbits is further increased to facilitate the photocatalytic reaction. During the reaction, defects and elemental doping result in the formation of unsaturated metals Zn^δ+^or Cu^δ+^ serving as the active center of the reaction, which facilitates the charge transfer from the ZnAl-LDH surface to CO_2_ to form COOH* intermediates and ultimately remove water to form CO, as illustrated in Figure 10. Our work reveals the intrinsic mechanisms of the coordinative effects of co-doped transition metal and oxygen defects, as well as metal defects, on effective photocatalytic CO_2_ reduction. This provides a new opportunity for the design of novel LDHs photocatalysts for the conversion of CO_2_.

## 3. Computational Method

All calculations are based on density functional theory utilizing the Vienna Ab initio Simulation Package (VASP) [43,44,45,46]. The exchange-correlation energy is evaluated utilizing Perdew–Burke–Ernzerhof (PBE) functional in the Generalized Gradient Approximation (GGA), and the core electron interaction is substituted with the Projector Augmented Wave (PAW) pseudopotential [47,48,49]. The van der Waals interaction is corrected by the DFT-D3 method [50,51,52] of Grimme and the spin polarization is considered [53,54]. To account for the strong electron correlation properties of Cu, the DFT + U methods [55] were employed with a U-value of 3.60 eV [56] for the Cu 3d state to consider the strong field Coulomb interaction of Cu local electrons [25,41,42,56,57]. The 4 × 4 × 1 Monkhorst–Pack mesh is employed to sample the k points in the Brillouin zone for structural relaxation, and the Plane Wave basis function is utilized to expand at 450 eV. To eliminate interaction between periodic units, the thickness of the vacuum layer is set to 15 Å, and a 3 × 3 × 1 periodic cell is established to avoid lateral interaction [58]. Calculations were carried out until the energy and force converged within 10^−6^ eV and 0.01 eV · Å^−1^, respectively [31,58]. To achieve the rapid convergence of the self-consistent field iterations, 0.1 eV Fermi smearing and Pulay mixing were utilized [39,40]. The energy band calculations of all three ZnAl-LDHs were performed using the hybrid functional HSE06 method [19]. The adsorption energies of adsorbates were determined by employing the formula [59,60]:(1)$${\Delta E}_{\mathrm{ads}}{=E}_{\left(a-s\right)}\;-\;E_{\left(s\right)}\;-\;E_{\left(a\right)}$$
where E_(a−s)_, E_(s)_, and E_(a)_ are the energies of the total adsorbate–substrate systems, isolated adsorbate, and isolated substrate, respectively. For each step of the reaction, the Gibbs free energy (G) [61,62] is computed using the following formula:(2)$${G=E}_{\mathrm{Total}}+\int C_{P}\mathrm{dT}+E_{\mathrm{ZPE}}-\;\mathrm{TS}$$
where E_Total_ represents total electron and ion energy, ∫C_P_dT represents enthalpy temperature correction, E_ZPE_ represents zero-point vibration energy correction, TS represents entropy contribution, T represents temperature (298.15 K), and S represents entropy [63,64].

The standard hydrogen electrode (SHE) model assumes that each step of the process will experience simultaneous proton transfer, and the electron pair will depend on the applied potential. Thus, the change in free energy concerning the initial state of CO_2_ gas above the empty surface can be described as follows:(3)$${\Delta G}_{{\mathrm{COOH}}^{*}}=G_{{\mathrm{COOH}}^{*}}+G_{H^{+}+e^{-}}\;-\;G_{*}-G_{{\mathrm{CO}}_{2}}-{2G}_{H^{+}+e^{-}}$$
(4)$${\Delta G}_{{\mathrm{CO}}^{*}}=G_{{\mathrm{CO}}^{*}}+G_{H_{2}O}-G_{*}-G_{{\mathrm{CO}}_{2}}-{2G}_{H^{+}+e^{-}}$$
(5)$$G_{H^{+}+e^{-}}=\frac{1}{2}G_{H_{2}}$$
where * represents the appropriate adsorption state on the catalyst’s surface, and “e” represents the fundamental charge.

Herein, the ZnAl-LDH system is subjected to ab initio molecular dynamics (AIMD) simulations utilizing a Nose–Hoover temperature-controlled NVT regularized synthesis with a time step of 1 fs. The initial configuration for the simulations is the DFT-optimized ZnAl-LDH, and the AIMD simulation time is increased to 10 ps at a temperature of 600 K to guarantee that the loaded system reaches the equilibrium state [65].

## 4. Conclusions

This work designs defect-rich ZnAl-LDHs photocatalysts by introducing oxygen and zinc defects and the incorporation of the transition metal copper. The stability, electronic structure, energy band structure, and possible usage in catalytic CO_2_ and their corresponding mechanisms have been examined by employing the DFT methods. The origins of the intriguing coordinative effects of defects and transition state metal co-doping of ZnAl-LDHs on the improvement of photocatalytic performances have been emphasized. The addition of V_Zn_, V_O_, and Cu to ZnAl-LDH significantly reduces the energy band gap and improves light absorption, and leads to the formation of the corresponding intermediate band (defective energy level/doping energy level), which can effectively suppress the recombination for creating photogenerated of electrons and holes and improve carrier mobility, facilitating electron transmission to the surface to take part in the catalytic reduction reaction. Furthermore, the CBM is lowered and the enhanced reduction capacity can drive a more efficient photocatalytic reduction of CO_2_ to CO.

Upon adding V_Zn_ and V_O_ to ZnAl-LDH, the thermal stability is improved and the Zn^δ+^-V_O_ active site is formed since the lattice is distorted. The stability of the defect structure is further improved after introducing Cu, and the Cu^δ+^-V_O_ complex is formed, serving as the active site, which could capture the photogenerated electrons more effectively, enhancing the bonding effect and facilitating the bonding with the reactant molecules. The emergence of Zn^δ+^ and Cu^δ+^ brings the energy level of the d-band center closer to the Fermi level, improving the adsorption and activation capacity of CO_2_ molecules and lowering the reaction barrier, which aids in the creation of intermediates. At the same time, the low-valent metal Zn^δ+^ and Cu^δ+^ can enhance CO selectivity by reducing CO adsorption and suppressing the HER reaction. Meanwhile, theoretical calculations show that the unsaturated complexes Zn^δ+^-V_O_ and Cu^δ+^-V_O_ form after the introduction of oxygen vacancies, metal vacancies, and metal Cu, in that order, and that their Zn-O and Cu-O bond lengths are consistent with the experimental values. DOS shows that the characteristic peaks of V_Zn_-ZnAl-LDH and Cu-V_Zn_-ZnAl-LDH in XPS correspond to Zn 4s orbitals and Cu 3d orbitals, respectively, which are electron-rich states that can be used as active sites for the reaction, effectively reducing CO_2_ to CO.

Defect engineering and element doping can significantly lower ZnAl-LDH charge carrier transport resistance, leading to more active sites and increased photocatalytic activity. Compared to pure ZnAl-LDH, V_Zn_-ZnAl-LDH can efficiently narrow the band gap and absorb light at longer wavelengths attributed to the V_O_ and V_Zn_, and the more negative CBM position can improve the selectivity for CO. Cu-V_Zn_-ZnAl-LDH has good thermal stability, the easiest reduction of CO_2_ to reduce CO, and efficient inhibition of hydrogen release, in addition to more effective collection of more visible light and selective enhancement benefits. This work explored the mechanism of CO_2_PR reaction by functionalized ZnAl-LDHs, which provides theoretical guidance for the design of novel LDH photocatalysts.

## Figures and Tables

**Figure 1 molecules-28-00738-f001:**
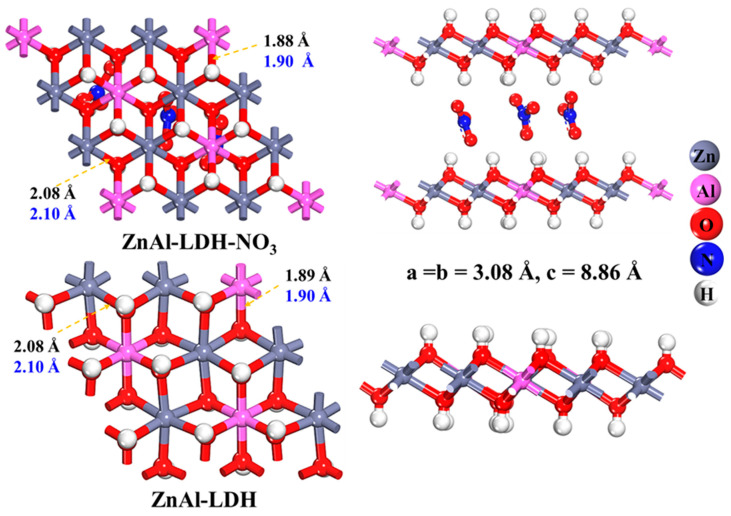
Geometric structures of ZnAl-LDH-NO_3_ and ZnAl-LDH optimized by the PBE + vdW method together with experimental values [31] in blue.

**Figure 2 molecules-28-00738-f002:**
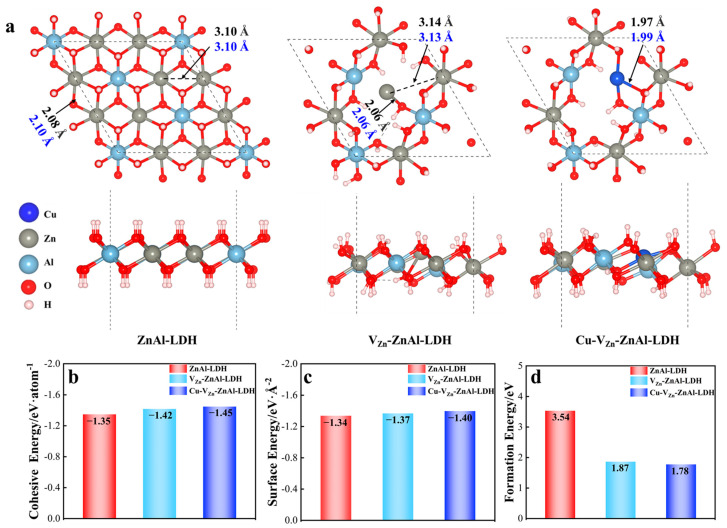
(**a**) Key geometrical parameters of three ZnAl-LDHs with DFT values in black and experimental values in blue. The stability is measured by (**b**) cohesive energy, (**c**) surface energy, and (**d**) formation energy.

**Figure 3 molecules-28-00738-f003:**
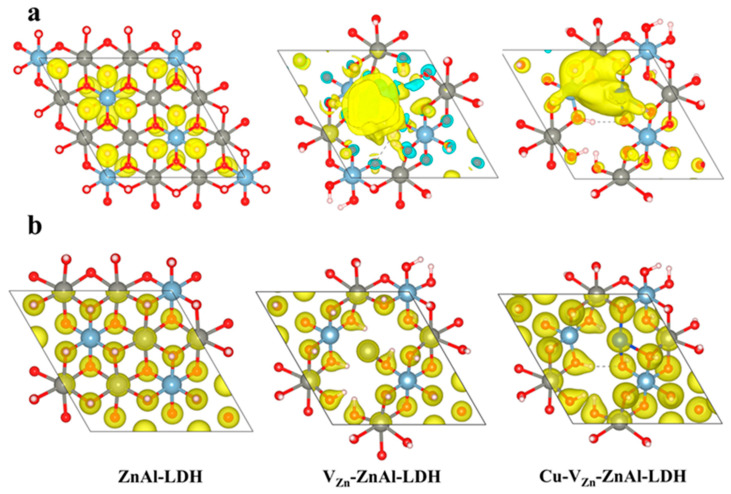
(**a**) The spin density and (**b**) the charge density of three ZnAl-LDHs.

**Figure 4 molecules-28-00738-f004:**
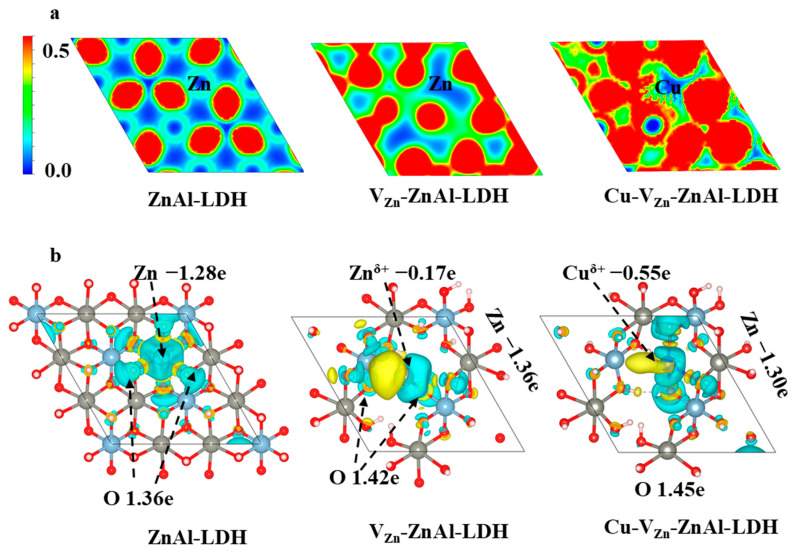
(**a**) Electron localization function (ELF) and (**b**) electron density difference (EDD) and Bader charges of the three ZnAl-LDHs.

**Figure 5 molecules-28-00738-f005:**
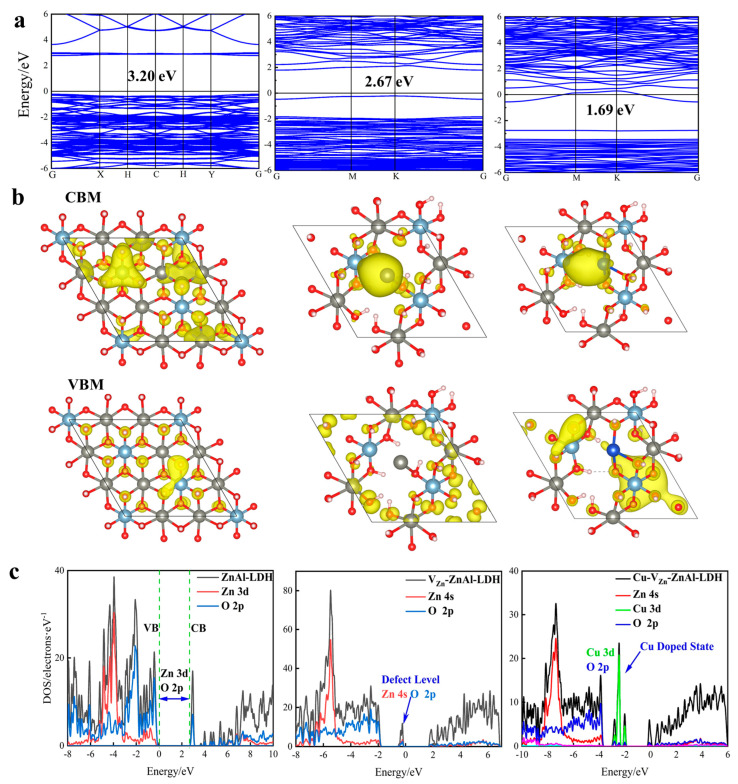
(**a**) Band structure, (**b**) charge density of VBM and CBM, and (**c**) related DOS of ZnAl-LDH (**left**), V_Zn_-ZnAl-LDH (**middle**), and Cu-V_Zn_-ZnAl-LDH (**right**).

**Figure 6 molecules-28-00738-f006:**
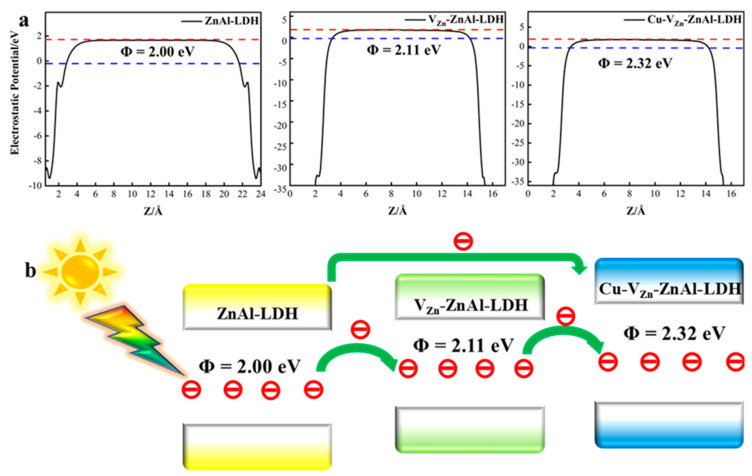
(**a**) Work function of the three ZnAl-LDHs where the red and blue lines represent the vaccum and the Fermi level, respectively; (**b**) electron transfer between ZnAl-LDHs.

**Figure 7 molecules-28-00738-f007:**
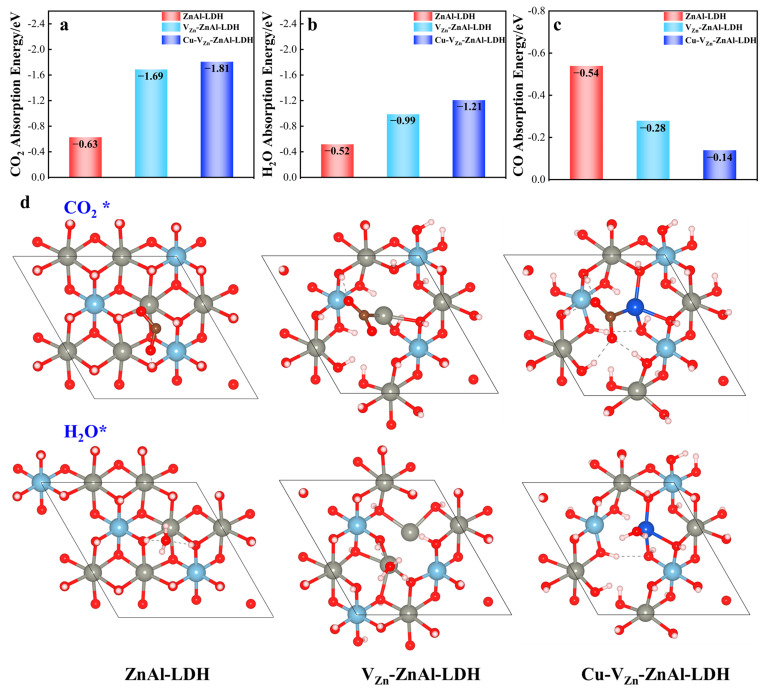
The adsorption energies of (**a**) CO_2_, (**b**) H_2_O, and (**c**) CO are absorbed on three ZnAl-LDHs, and (**d**) related structures for CO_2_ (**up**) and H_2_O (**down**) adsorption (The “*” represents the appropriate adsorption state on the catalyst’s surface).

**Figure 8 molecules-28-00738-f008:**
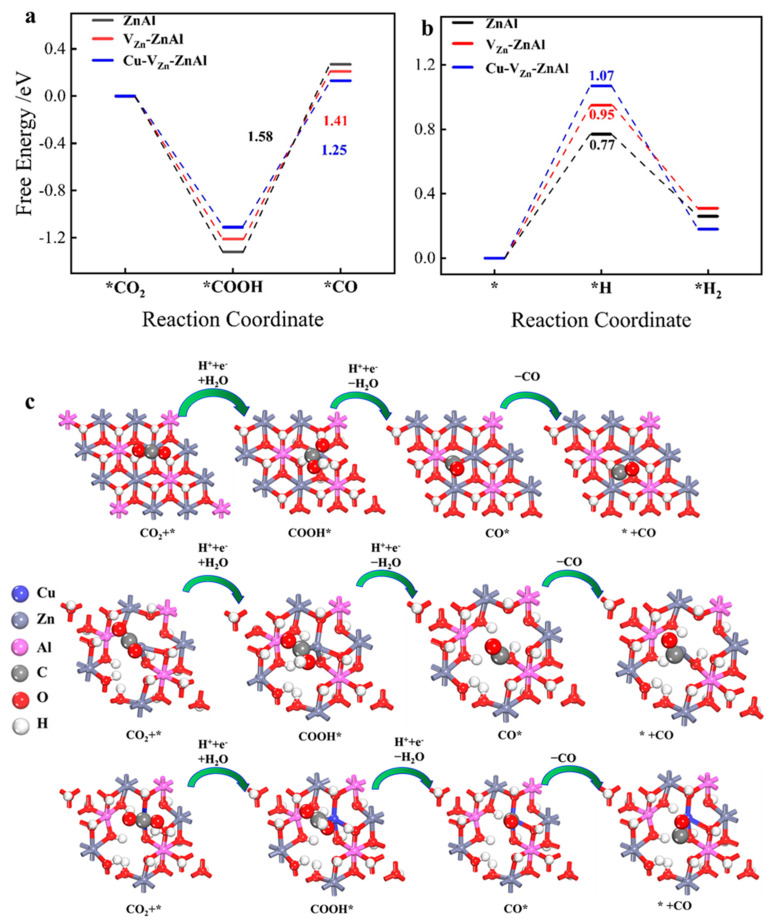
The Gibbs free energy diagrams of (**a**) CO_2_ reduction, (**b**) H_2_ evolution, and (**c**) intermediates during CO_2_ reduction to CO over ZnAl-LDH, V_Zn_-ZnAl-LDH, and Cu-V_Zn_-ZnAl-LDH.

**Figure 9 molecules-28-00738-f009:**
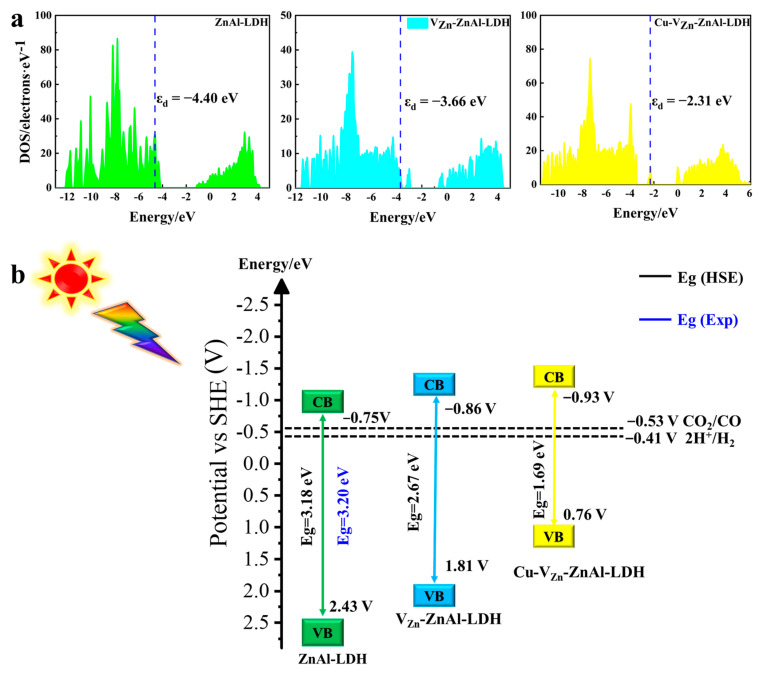
(**a**) The d-band center positions and (**b**) the CB and VB potentials of the three ZnAl-LDHs.

**Figure 10 molecules-28-00738-f010:**
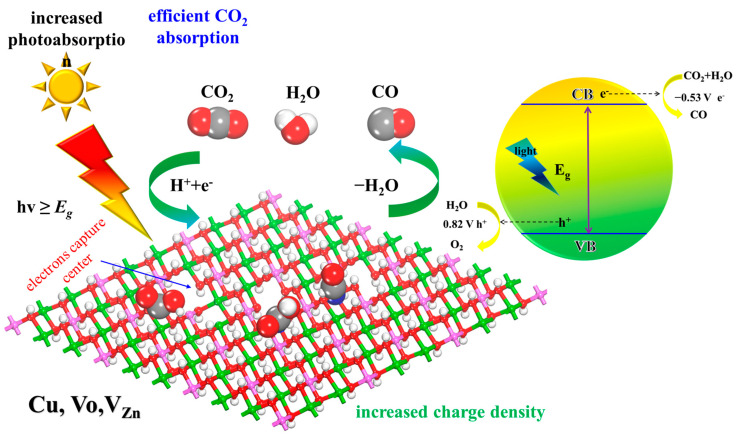
Illustration of the photocatalytic reduction mechanism of CO_2_. Atomic labels Al, Zn, Cu, C, O, and H are pink, green, blue, gray, red, and white, respectively.

## Data Availability

The data presented in this study are available in Appendix A.

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
