# Peer review of "Functional Regulation of ZnAl-LDHs and Mechanism of Photocatalytic Reduction of CO2: A DFT Study"

_molecules, 2023, doi:10.3390/molecules28020738_

Round 1

Reviewer 1 Report

This article can arouse the reader's interest. It can be published on Molecules after a revision.

Q1: Please explain the superiority of using different ZnAl-LDHs doped photocatalysts for photocatalytic reduction of CO2.

Q2: Please compared your theoretical results with real experimental date to confirm the photocatalytic reduction of CO2 using the ZnAl-LDHs doped photcatalysts.

Q3: On the first page, lines 43 to 45, only the limitations of the current photocatalysts are given, please give the relevant literature support in recent years.

Q4: Please refer to recent published work focus on the photocatalytic reduction of CO2 using different element-doped ZnAl-LDHs photcatalysts.

Q5: Page 2, line 87, duplicate references; page 3, lines 115-116, equations (3), (4) and (5) do not mention the "U" given in the text, please double check and correct.

Q6: Please explain the meaning of total energies in E(a-s), E(s), and E(a) are the total energies of the adsorbed molecules and ZnAl-LDHs com-plexes, ZnAl-LDHs, lines 101 to 102 on page 3.

Q7: Lines 145-146, 211-212, and 266-267, the data given in the text do not agree with the data in the figure, please verify and correct.

Q8: In Figure 8c, the names of intermediates are not given below some diagrams, please check carefully

Reviewer 2 Report

The manuscript is written and presented well and is well in context of current research interests. The manuscript can be published after minor corrections.

1. Atoms in the figures should be labeled or color code should be mentioned.

2. Font in all the figures should be uniform.

3. Manuscript should be thoroughly checked for English corrections.

Reviewer 3 Report

This manuscript describes the mechanism of enhanced photoctatalytic CO2 reduction over the respective zinc defect, oxygen decfect, and Cu doped ZnAl-LDH based on DFT, which is the state-of-the-art tool for understanding the photocatalytic mechanism. The results are well-presented and discussed.  This paper can provide a paradigmatic approach to study the underlying mechanism of defect and dopant engineering of LDH-baed photocatalysts for CO2 reduction, rendering it not only important in the area of theoretical chemistry but also in the lab-baesd studies. Therefore, I recommend the acceptance of the manuscript in the current form.

Author Response

Thank you to the reviewers for their positive support and agreement for acceptance of our manuscript.

Reviewer 4 Report

The authors design defect-rich ZnAl-LDH photocatalysts, introducing oxygen and zinc defects and the incorporation of copper, the transition metal. The stability, electronic structure, energy band structure and possible use in catalytic CO2 and its corresponding mechanisms were examined using DFT methods. The methodology and results appear to be coherent. I recommend this paper for publication after a minor revision. Please, find below my comments and suggestions:

1)    The abstract and the paper has very long sentences, some exceed three lines. I suggest a small English review to shorten all long sentences. It is not recommended to have sentences longer than two lines even if the language is clear.

2)    Vo should be defined upon first use in the abstract.

3)    The introduction focuses on photoreduction of CO2 with only 3 references, one of which is a paper (ref, 1-3). Please add a general introduction on the general chemistry of CO2, the importance of CO2 capture for climate change and for synthesis of value-added products and cite the following reviews:

A comprehensive review on different approaches for CO2 utilization and conversion pathways; https://doi.org/10.1016/j.ces.2021.116515

Introductory Chapter: An Outline of Carbon Dioxide Chemistry, Uses and Technology

DOI: 10.5772/intechopen.79461

4)    In figure 2a, place the legend of the colors of the atoms for a better understanding of the geometry.

5)    In de figure 4b, the three ZnAl-LDHs is the same order the figure 4a? Please, put in the legend for better understanding.
